# Classification of Liquid Ingress in GFRP Honeycomb Based on One-Dimension Sequential Model Using THz-TDS

**DOI:** 10.3390/s23031149

**Published:** 2023-01-19

**Authors:** Xiaohui Xu, Wenjun Huo, Fei Li, Hongbin Zhou

**Affiliations:** 1School of Armament Science and Technology, Xi’an Technological University, Xi’an 710064, China; 2School of Mechatronic Engineering, Xi’an Technological University, Xi’an 710064, China; 3School of Equipment Management and UAV Engineering, Air Force Engineering University, Xi’an 710043, China

**Keywords:** GFRP, liquid ingress, defects classification, THz-TDS, one-dimension sequential model

## Abstract

Honeycomb structure composites are taking an increasing proportion in aircraft manufacturing because of their high strength-to-weight ratio, good fatigue resistance, and low manufacturing cost. However, the hollow structure is very prone to liquid ingress. Here, we report a fast and automatic classification approach for water, alcohol, and oil filled in glass fiber reinforced polymer (GFRP) honeycomb structures through terahertz time-domain spectroscopy (THz-TDS). We propose an improved one-dimensional convolutional neural network (1D-CNN) model, and compared it with long short-term memory (LSTM) and ordinary 1D-CNN models, which are classification networks based on one dimension sequenced signals. The automated liquid classification results show that the LSTM model has the best performance for the time-domain signals, while the improved 1D-CNN model performed best for the frequency-domain signals.

## 1. Introduction

Light-weight and high-strength GFRP honeycomb structures are popular in the aerospace industry. However, volatile environment conditions and peeling of surface coatings during service can easily trigger liquid ingress. In most cases, a small amount of liquid ingress is tolerable. Yet, a few hydraulic oils can cause serious mechanical failure in some critical components. In the case of wave-transparent GFRP honeycomb structures, such as radomes, water ingress will severely degrade wave-transparent performance. Therefore, the detection and classification of liquid ingress in honeycomb structures becomes an essential step in the manufacturing process and during service [1,2,3,4].

Currently, the quality control and flaw detection of GFRP are mainly conducted using ultrasonic methods [5]. However, the couplant is indispensable, hindering ultrasonic methods to non-waterproof structures. X-ray is capable of tomography scan, providing high-resolution images. However, this method is harmful to human heath, and the hardware is very expensive [6]. Infrared thermography is a relatively new technology with the advantages of fast and non-contact detection, human independence, high spatial resolution, and high acquisition rate. Recently, several infrared stimuluses have been developed, such as flash lights and ultrasonic vibration [7], eddy current [8], and laser spot [9]. However, infrared technology is not suitable for thick materials with poor thermal conductivity, such as GFRP. In addition, for the low infrared emissivity and high reflectivity inspection surface, it is necessary to carry out painting or other surface treatment. Therefore, all the above inspection techniques have drawbacks in the inspection of composite materials.

Terahertz (THz) waves enable non-destructive testing and evaluation (NDT&E) and thus have gained increasing attention recently. As a typical case, non-polar materials were successfully inspected using THz-TDS [10]. Moreover, terahertz spectral signals that carry abundant information can be used for defect feature extraction and classification. On the other hand, recurrent neural network (RNN) and 1D-CNN are expert at handling such sequence signals. Specifically, RNN is capable of memorizing forward information, which outperforms at tasks such as natural language processing and sequential data prediction. However, gradient disappearance or explosion may occur when RNN deals with long-time sequential data. To solve this problem, LSTM structure is proposed and improves the ability of rather long-distance information extraction. CNN has been leveraged for extracting features from two-dimensional images. By altering the convolutional dimension, 1D-CNN could also extract features from sequential data. Compared with LSTM-RNN, 1D-CNN has advantages of faster training, parallel calculation, and alike performance.

Recently, defect identification or classification has been reported. Fuzhou Shen et al. presented an open-source mobile multispectral imaging system, tested the influence of the utilization of LEDs on the multispectral image, and designed image-processing algorithms to correct this influence [11]. Tao et al. compared LSTM with artificial feed-forward neural network on inspection of non-planar carbon fiber reinforced plastic samples using pulsed thermography and showed that LSTM was more accurate in handling time-dependent information compared to the artificial feed-forward neural network model [12]. Duan et al. utilized a neural network in infrared thermography to classify defects in materials and showed that the model is precise and can be generalized [13]. Guo et al. proposed a novel detection baseline model based on a fully convolution network and gated recurrent unit to classify ultrasonic signals from flawed three-dimensional braided composite specimens with debonding defects. Experimental results based on an in-house dataset showed that the proposed model performs very well against all baselines [14]. Wang et al. used ordinary CNN to realize the classification of liquid contraband [15]. Marani et al. introduced a novel methodology for the automatic analysis of thermal signals resulting from the application of pulsed thermography and demonstrated that the proposed approach has better classification performance over other approaches in terms of the reduction of missing predictions of defective classes [16]. One-dimensional sequential model has been proven to have good performance in one-dimensional data processing, classification, and prediction. However, unpreprocessed signals and traditional network model structure limit the classification accuracy and speed for specific detection signals.

In this work, we extend sequential data processing model into classification of liquid ingress in GFRP honeycomb structure. We will focus on three types of liquid ingresses, water, oil, and alcohol. We will first introduce the acquisition of time-domain signals from three defective regions and one non-defective region as datasets. Then, we will study the smoothing of the signals by using an improved wavelet denoising algorithm. Finally we will establish an improved 1D-CNN to extract features and classify the liquid ingress. LSTM and 1D-CNN are compared with the improved network to verify its performance.

The main contributions of this work are as follows:We construct three THz datasets of liquid ingress, including oil, water and alcohol, as well as a dataset of non-defective GFRP honeycomb structure. Through the optical parameter calculation model, we obtain refractive index, absorption coefficient, dielectric constant, and other spectral parameters, which are useful for the classification and detection of liquid ingress.We propose an improved hard threshold wavelet denoising algorithm, which facilitates feature extraction of the neural network. By using a non-linear smooth decay function, the noise is effectively removed while preventing local distortion of the signal. The noise reduction performance can be optimized by applying different threshold values at different time stages of the signal.We develop an improved 1D-CNN model for the identification and classification of liquid ingress based on THz signals. According to THz signal characteristics, convolutional skip-connection module and fractional step convolutions are added to extract the signal features. The improved 1D-CNN performs well in the classification of liquid ingress.

## 2. Signal Noise Reduction Processing

The noise-reduction process can reduce the waveform variations of the terahertz time-domain signals, and thus is beneficial for the feature extraction of the neural network. In this work, based on wavelet noise reduction algorithm, the traditional threshold processing method is improved, since a smoothing function is introduced to effectively eliminate noise and prevent local signal distortion in the meantime. The specific steps are as follows.

We firstly performed wavelet decomposition on the one-dimensional signal data, i.e., after shifting τ for a basic wavelet function, and then made an inner product with the signal y(x) to be analyzed at different scales *a*, which can be expressed as
(1)$$WT_{f}\left(a,\tau \right)=\left[y\left(x\right),\psi_{a,b}\left(x\right)\right]=\frac{1}{\sqrt{a}}\int y\left(x\right)\psi \left(\frac{x-b}{a}\right)dx\;,$$
where $\psi_{a,b}\left(x\right)=\psi \left((x-b)/a\right)/\sqrt{a}$ is a set of wavelet basis function obtained from the wavelet basis ψ(x) through scaling and translation, *a* is the scale factor for the basic wavelet function φ(x), and *b* is the translation factor reflecting the size of shift. The duration time of the wavelet at different scales widens when $\psi_{a,b}\left(x\right)$ increases, whereas the amplitude decreases in inverse proportion, but the wave shape remains the same. It can be seen that the wavelet transform proposes a changing time window. When low-frequency signals are needed, the long-time window should be adopted precisely. However, it has been shown that a short-time window is suitable for the high-frequency signals [17,18]. Depending on the frequency range of terahertz signals, low frequency information is needed. The original signal can be decomposed at different scales to obtain its information at different scales. The original signal can be obtained by wavelet inversion of wavelet coefficients,
(2)$$y\left(x\right)=\frac{1}{C_{\psi }}\int_{0}^{\infty }\frac{1}{a^{2}}da\int WT_{f}\left(a,b\right)\psi_{a,b}\left(t\right)dt\;.$$

For better implementation, we utilized discrete wavelet transform (DWT) to reduce the redundancy of wavelet transform. In the discretization, the scale is discretized by power series, i.e., by setting $a=a_{0}^{j}$, and the discrete binary wavelet becomes:(3)$$\psi_{j,k}\left(x\right)=\frac{1}{\sqrt{2^{j}}}\psi \left(\frac{x}{2^{j}}-k\right)\;.$$

Thus, the discrete wavelet variation DWT can be written as
(4)$$Y\left(j,k\right)=DWT_{f}\left(j,k\right)=\int y\left(x\right)\psi_{j,k}\left(x\right)dx\;.$$

According to the basic steps of the previous signal wavelet noise reduction, once the wavelet decomposition coefficients are obtained, the wavelet coefficients can be processed by selecting a suitable threshold value. There are currently two basic threshold methods: the soft-threshold method and the hard-threshold method. The hard-threshold method can have a better signal-to-noise ratio, but may distort the signals. To overcome this problem, we develop a modified hard-threshold method, which is expressed as:(5)$$Y_{thr}=\left\{\begin{array}{cc} T\cdot \frac{\exp\left(a|Y|/T\right)-1}{\exp\left(a\right)-1}\cdot \mathrm{Sign}\left(Y\right) & |Y|<T\;\\ Y & |Y|\geq T \end{array}\right.\;,$$
where *T* is the threshold. By using a non-linear smooth decay function, the noise is effectively removed while preventing local distortion of the signal.

Since the low frequency noise is strong in the initial stage, but disappears as the time length increases, the noise reduction performance can be optimized by applying different threshold values at different time stages of the signal. After that, wavelet inversion reconstruction is carried out to obtain the noise-reduced signals.

Figure 1 depicts a typical terahertz time-domain signal before and after noise reduction. Results show that the signal noise is significantly reduced, while the signal waveform stays almost the same.

## 3. Neural Network Structure

### 3.1. 1D-CNN

Both terahertz time-domain and frequency-domain signals are one-dimensional sequence data, thus the 1D-CNN can be used to extract the features [19,20]. 1D-CNN usually uses the structure of convolutional layer and pooling layer hierarchy. After each convolutional layer, the data are compressed in a lossy way, which is referred to as down sampling. 3×1 convolution kernel and a convolution layer depth of 6 are used in this network. Finally, a fully connected layer and sigmoid function is set for prediction result output. The 1D-CNN structure is illustrated in Figure 2.

### 3.2. 1D-CSNN

It has been shown that the ordinary 1D-CNN has the ability of extracting deep, effective, and robust features from sequence signals via one-dimensional convolutional layers. However, the cascaded approach of ordinary CNNs has some limitations because the outputs of shallow and intermediate layers also contain some potential feature information for more accurate classification results. Therefore, based on the ordinary 1D-CNN, we develop improved structures and we refer to them as one-dimensional convolutional skip-connected networks (1D-CSNNs). As illustrated by Figure 3, we add a convolutional skip-connection module to the ordinary 1D-CNN, which can extract the shallow and middle layer features and thus effectively avoid features loss.

Convolutional skip-connections play an important role in both forward and backward propagations. On one hand, compared with traditional skip-connection, each convolutional skip-connection introduces a small block of convolution compared to a conventional cascaded approach, which slightly adjusts the feature mapping extracted during forward propagation. On the other hand, the convolutional skip-connections alleviate gradient disappearance in back propagation as the gradients are passed directly through the skip-connections. Specifically, we use a small 3×1 convolutional kernel and activation function ReLU as the first layer. For blocks 2 to 3, we apply some series of fractional step convolutions (FSC) to all convolutional layers. Given that batch normalization (BN) is often used to offset internal covariate shifts [21], BN helps bridge the gap between FSC and parametric convolution. For the activation function, we use PReLU, which can be expressed as
(6)PReLU=max(x,0)+amin(0,x). Here, *x* denotes the input signal and the parameter *a* is learnable, which can effectively alleviate the“dead feature” in zero gradient [22]. The prediction result outputs via a fully connected layer and sigmoid function.

### 3.3. LSTM

We also compared the 1D-CSNN with long short-term memory recurrent neural network (LSTM-RNN, shortened as LSTM), which was used by some of the authors [23]. LSTM is a special structured RNN for processing longtime signals, which performs better than the normal RNN and solves the problem of gradient disappearance and gradient explosion during the training of longtime data. The structural design focuses on adding a fully-connected layer after the output layer of the LSTM, and classification results output via the sigmoid function.

## 4. Sample Preparation and Experimental Setup

### 4.1. GFRP Honeycomb Sandwich Sample with Liquid Ingress

An unsealed GFRP honeycomb sandwich sample was used for the terahertz inspection experiments. Figure 4a shows the sample with four regions filled with liquid ingresses. The surface skin was made of glass fibre fabric 3218 epoxy resin prepreg curing, and both the front and back surface skins have four layers, and each layer has the thickness of about 0.22 mm. The honeycomb layer was made of hexagonal Nomex paper honeycomb with 20 mm thick and 3 mm long of a single grid. Before the surface skin is mounted, four different regions of the honeycomb sandwich are filled with 10 mm thick water, oil, and alcohol, respectively, using a syringe. Finally, the top surface skin is mounted and sealed, as shown in Figure 4b.

### 4.2. Experimental Setup

We measured the terahertz reflection spectra from the as-prepared sample using a THz-TDS system combined with a robot arm, as shown in Figure 5. The robot arm works as a terahertz probe carrier, and the scanning path of the arm is planned by off-line programming. During the scanning, the continuous movement of the arm is synchronized with the real-time acquisition of terahertz signals. The THz-TDS system utilizes a fibre optic femtosecond laser (wavelength 1550 nm, pulse width 90 fs) and a fibre-coupled photoconductive antenna. The system has a spectral width of about 2 THz and a dynamic range of around 60 dB.

## 5. Experimental Results Analysis

### 5.1. Data Set Construction

The terahertz time-domain spectral signals from the three liquid ingresses and a non-defective region were selected to construct the data set. These signals were labelled as water, oil, alcohol, and non-defective, correspondingly. After filtering, 23,682 signals were obtained, within which 3340, 3733, 3157, and 13,452 signals were labelled with water, oil, alcohol, and non-defective, respectively. To improve generalizability of the training, 3817 signals from the non-destructive signals were randomly extracted to build the data set. The data set made of 14,047 signals were divided into a training set, a validation set, and a test set with a ratio of 7:2:1, according to the principle of mutually exclusive identical distribution. The detailed information of the data set is listed in Table 1.

### 5.2. Interpretability Analysis

To analyze the interpretability of liquid ingress classification by one-dimensional sequence model, we selected signals of each liquid ingress area, which are shown in Figure 6.

Figure 6a shows that the terahertz reflection signals of water and non-defective area are easily distinguishable, whereas the signals of alcohol and oil are similar. The signal of water reaches the maximum value around 11.25 ps, whereas that of non-defective area reach the minimum value around 10.75 ps. Thus, we infer that the accuracy of distinguishing water and non-defect will be relatively high. On the other hand, the amplitude, the peak occurrence time, and peak value of the alcohol signal and oil signal are close to each other. Therefore, we speculate that the accuracy of distinguishing oil and alcohol will be relatively low.

Figure 6b depicts the frequency-domain reflection signals. The reflection peak represents weak absorption, and, on the contrary, the reflection trough represents strong absorption due to the hydrogen bond of the object molecule resonates with the terahertz wave at this frequency. With regard to the four signals, their strong absorption frequency (SAF) and weak absorption frequency (WAF) are different from each other. To be more specific, both SAF and WAF of water signal and non-defective signal are far from each other; both SAF and WAF of oil signal are almost equal to these of alcohol signal, but their corresponding amplitudes at WAF differ remarkably. Therefore, the classification of these frequency-domain signals will be appreciable.

### 5.3. Training Results

The cross-entropy loss function was set as the loss function, and the adaptive moment estimation (Adam) gradient fastest descent method was chosen for the gradient descent algorithm. The computer hardware configurations mainly included an Intel Core i7-12700k processor with 32 GB of memory and an NVIDA RTX 3070 graphics card. The deep learning framework was Pytorch. The labeled training set was fed into the three neural networks for training, and some of the hyperparameters of the networks were adjusted according to the validation result from the validation set. After that, the learning rate was set to 0.0001, and the batch size was set to 128. The changes of the loss and accuracy during training and validation are shown in Figure 7. Loss indicates the loss value of training set, and val_loss indicates the loss value of verification set. Acc and val_acc are the same.

Results show that, as epoch increases, the training losses (blue curves) of all the three networks decrease, and meanwhile the accuracy (red curves) keeps increasing until saturates. These behaviors suggest that all the three networks perform well. The accuracy curves of the three are stabilized around 0.95 near 40 iterations, i.e., convergence is achieved. However, the smaller loss curve fluctuation of LSTM than 1D-CSNN and 1D-CNN indicates that the weights of LSTM could update toward the optimized direction of the objective function, whereas the gradient descending directions of 1D-CSNN and 1D-CNN are not certain. Therefore, LSTM performs better than 1D-CSNN and 1D-CNN in extracting time-domain signals.

We also fed the frequency-domain signal dataset, which were obtained through fast Fourier transform on the time-domain dataset to the three networks. The loss and accuracy curves are shown in Figure 8. Results also show an overall decreasing loss and an increasing accuracy, indicating that the learning is also satisfactory. However, notable fluctuations appear in curves of LSTM and 1D-CNN, whereas curves from 1D-CSNN are stable. It can be concluded that 1D-CSNN works best for the frequency-domain signals.

Comparing the time-domain and frequency-domain data training of the three networks mentioned above, all of them could accomplish the task. LSTM has the best performance when handling time-domain signals, while 1D-CSNN is more suitable for frequency-domain signals.

### 5.4. Classification Results

We carried out THz-TDS inspection on the sample (point by point at a step of 0.5 mm) in reflection mode to obtain untrained time-domain signals. These signals were used to classify by the three trained network. The results are visualized using pseudo color maps in Figure 9, and the confusion matrices are shown in Figure 10 (the labels 0, 1, 2, and 3 in the confusion matrices correspond to non-defective, oil, alcohol, and water, respectively).

Results show that the classification accuracy of the non-defective region is the highest in both time and frequency-domain signals for all three networks. The reason is that without the interference of impurity, the non-defective signals are more stable and the waveform structure is much similar. These traits make it easy for neural networks to learn or extract signal features leading to high accuracy. However, the classification accuracy of alcohol and water is lower than oil. This is because the alcohol used here is a mixture of ethyl alcohol and water, making the signals from alcohol and water differentiate inconspicuously. Figure 9 and Figure 10 show that water ingress in the bottom right corner is incorrectly classified into alcohol. The possibility of actual water ingress misjudged as alcohol is relatively high, and the max reaches 0.18. However, the possibility of actual alcohol ingress misjudged as water is comparatively lower, and the max is only 0.05. This is because water in alcohol will strongly influence the classification accuracy.

Table 2 shows the evaluations of three sequential models using time-domain and frequency-domain signals. As for the single category prediction, LSTM has the highest precision of predicting oil. Compared with 1D-CNN, the proposed 1D-CSNN has a higher precision of every category and its performance is close to LSTM. F1-score is a comprehensive index of benchmarking networks, and LSTM outperforms when handling the time domain signals, while 1D-CSNN outperforms for the frequency domain signals. The F1-score of 1D-CSNN in frequency domain signals is 0.02 higher than that of LSTM in time domain signals. Meanwhile, the time cost of 1D-CSNN is lower than LSTM. In other words, the proposed 1D-CSNN not only preserves the high efficiency of CNN, but also has highly-improved performance on processing sequence signals.

LSTM and 1D-CNN have different performance for different one-dimensional sequence signal data. Time-domain signal is sequential and smooth, whereas frequency-domain signal contains notable numeric fluctuations, such as absorption peak, etc. Therefore, LSTM, which is built for time series data learning, excels at time-domain signals, and 1D-CSNN, which possess the prominent ability of feature extraction, stands out from handling frequency-domain signals.

## 6. Concluding Remarks

In conclusion, we have investigated the classification of liquid ingress in GFRP based on one-dimension sequential model using THz-TDS. We have compared the classification performance of three networks, LSTM, 1D-CSNN, and 1D-CNN, using both terahertz time-domain and frequency-domain signal data. Experimental results have shown that all the three networks can achieve classification of three different liquid ingress and non-defective region by learning time or frequency-domain data with a high accuracy over 85% and fast speed shorter than 8.31 s. Depending on the type of data, the learning ability of the networks differs. LSTM performs best for time-domain data, and 1D-CSNN is good at frequency-domain data. However, the improved network 1D-CSNN proposed in this work is close to LSTM in performance while faster than LSTM in speed.

One shortcoming of this experiment is that only the reflective scan mode is used, and the optical model is complex, making it difficult to calculate optical parameters, such as refractive index, absorption coefficient, and dielectric constant. This is because the absorption of terahertz waves by water is quite large, and terahertz waves become very weak after penetrating the sample in the transmission mode. However, the transmission mode provides more information about the optical parameters, which can be of great help to improve the accuracy of the classification of defects in composites. In the future, we intend to investigate the classification of solid defects in composites in transmission mode by adopting high-power terahertz sources and highly sensitive detectors. We expect that by introducing more optical parameters, better classification results can be obtained.

## Figures and Tables

**Figure 1 sensors-23-01149-f001:**
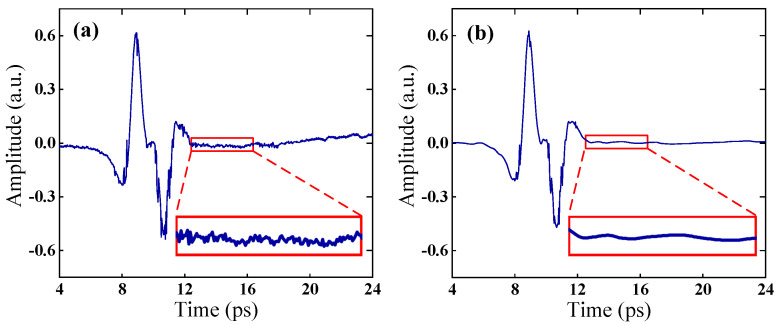
(**a**) Original terahertz time–domain signal, (**b**) denoised signal based on improved wavelet denoising method. Insets show a zoomed region.

**Figure 2 sensors-23-01149-f002:**
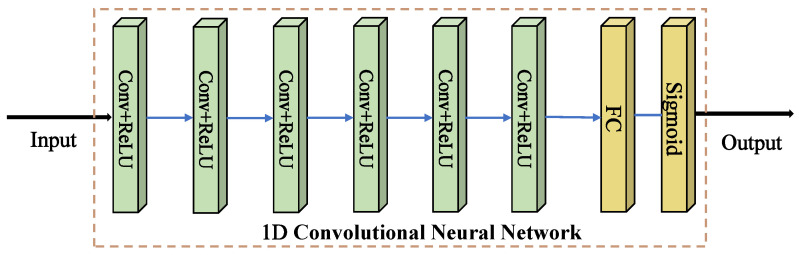
Schematics of the architecture of ordinary 1D-CNN, which is composed of convolution layers and pooling layers with the capability of feature extraction.

**Figure 3 sensors-23-01149-f003:**
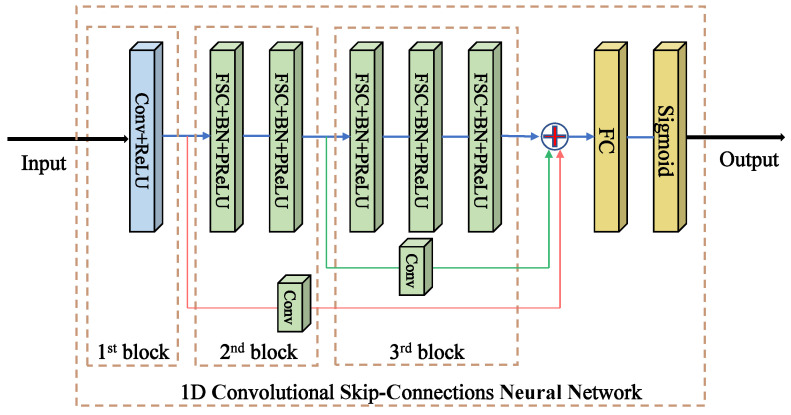
Schematic of the architecture of 1D-CSNN, which contains three blocks. Each block has a skip-connection line with the FC layer providing potential feature information to output.

**Figure 4 sensors-23-01149-f004:**
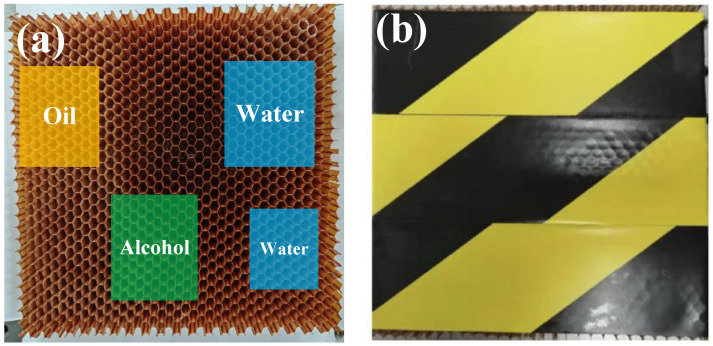
Pictures of the GFRP honeycomb sandwich structure with liquid ingress (**a**) before and (**b**) after the top surface skin is mounted and sealed.

**Figure 5 sensors-23-01149-f005:**
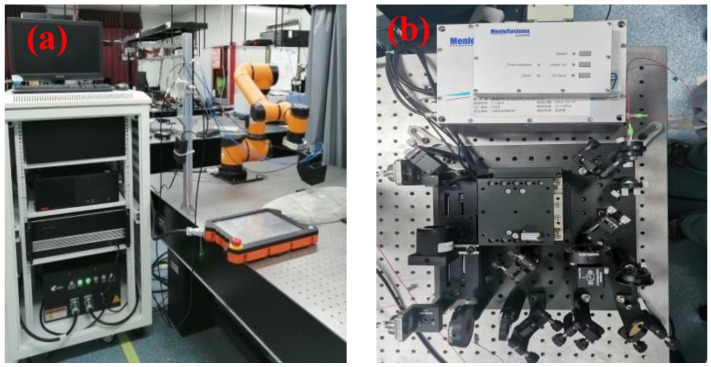
(**a**) Optical fiber coupled THz-TDS system combined with a robot arm for multi-degree-of-freedom manipulator scanning. (**b**) Zoom of the THz-TDS system.

**Figure 6 sensors-23-01149-f006:**
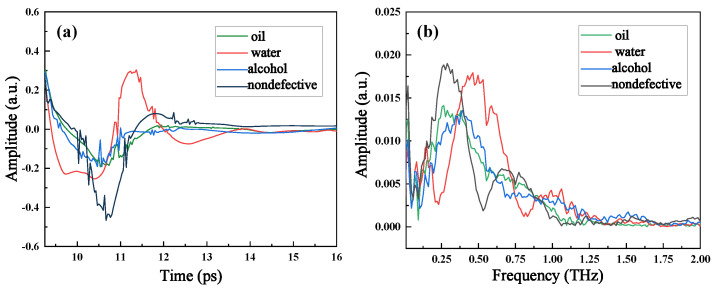
Terahertz signals of each liquid ingress in (**a**) time-domain in 9.25–15 ps window and (**b**) frequency-domain in 0–2 THz window.

**Figure 7 sensors-23-01149-f007:**
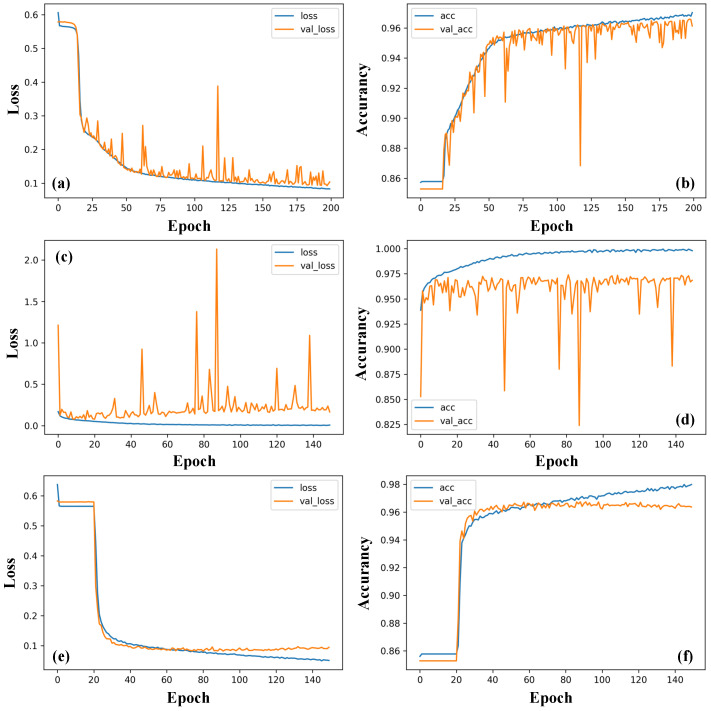
Training loss and accuracy curve of (**a**,**b**) 1D-CSNN, (**c**,**d**) 1D-CNN, and (**e**,**f**) LSTM based on time-domain signals.

**Figure 8 sensors-23-01149-f008:**
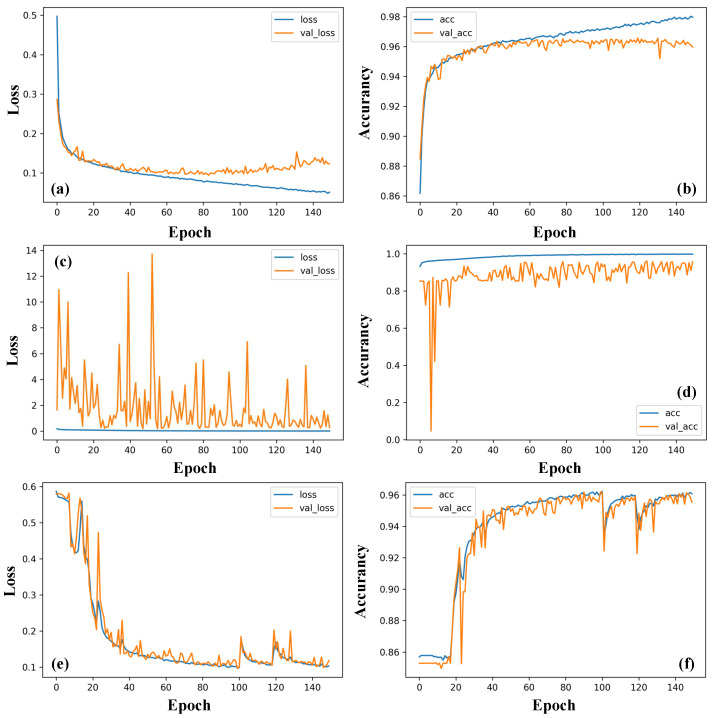
Training loss and accuracy curves of (**a**,**b**) 1D-CSNN, (**c**,**d**) 1D-CNN, and (**e**,**f**) LSTM using frequency-domain signal data.

**Figure 9 sensors-23-01149-f009:**
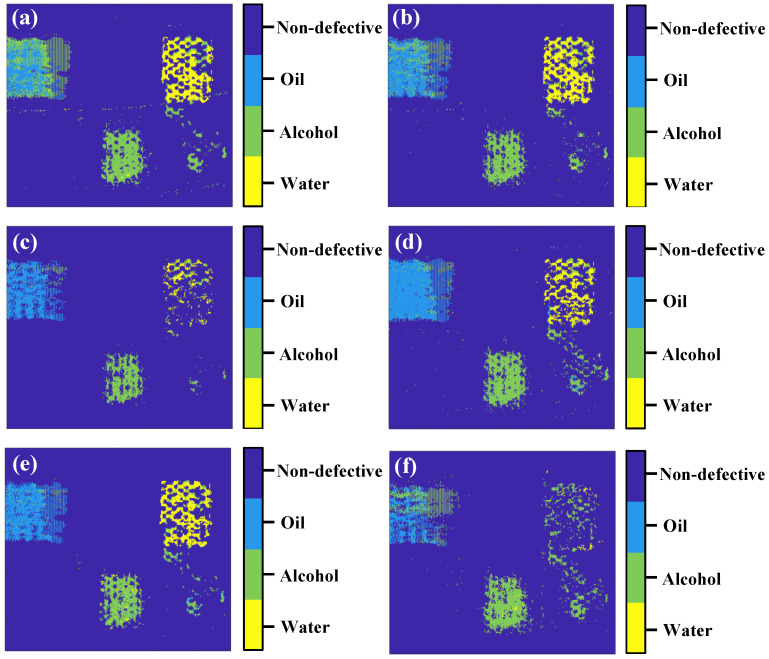
The classification results of three one-dimensional sequence models: (**a**,**b**) 1D-CSNN, (**c**,**d**) 1D-CNN, and (**e**,**f**) LSTM. (**a**,**c**,**e**) are obtained based on time-domain signals, and (**b**,**d**,**f**) are based on frequency-domain signals.

**Figure 10 sensors-23-01149-f010:**
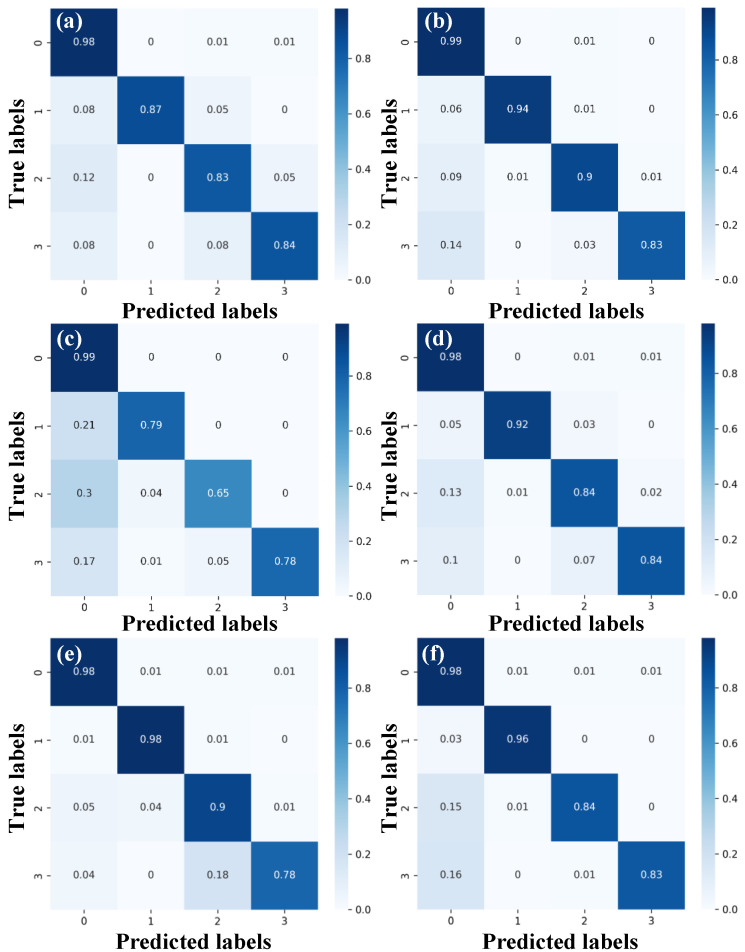
The confusion matrices of three one-dimensional sequence models: (**a**,**b**) 1D-CSNN, (**c**,**d**) 1D-CNN, and (**e**,**f**) LSTM. (**a**,**c**,**e**) are obtained based on time-domain signals, and (**b**,**d**,**f**) are based on frequency-domain signals.

**Table 1 sensors-23-01149-t001:** Sample sizes of each label.

Label	Water	Oil	Alcohol	Non-Defective
Training set	2338	2614	2209	2857
Validation set	668	746	631	622
Test set	334	373	315	338

**Table 2 sensors-23-01149-t002:** Evaluations of three sequential models using time-domain and frequency-domain signals. Bold values indicate that the evaluation results are the best in each category.

Model	Types	Precision		F1-Score		Time(s)	
		TD	FD	TD	FD	TD	FD
1D-CSNN	Non-defective	0.98	**0.99**	0.88	**0.93**	6.22	**5.94**
	Oil	0.87	0.94				
	Alcohol	0.83	**0.9**				
	Water	**0.84**	0.83				
1D-CNN	Non-defective	**0.99**	0.98	0.85	0.9	**6.05**	6.21
	Oil	0.79	0.92				
	Alcohol	0.65	0.84				
	Water	0.78	**0.84**				
LSTM	Non-defective	0.98	0.98	**0.91**	0.88	8.31	7.95
	Oil	**0.98**	**0.96**				
	Alcohol	**0.9**	0.84				
	Water	0.78	0.83				

## Data Availability

The experimental data are available locally.

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
