# Peer review of "Classification of Liquid Ingress in GFRP Honeycomb Based on One-Dimension Sequential Model Using THz-TDS"

_sensors, 2023, doi:10.3390/s23031149_

Round 1

Reviewer 1 Report

In this paper, an improved one-dimensional convolutional neural networks model was proposed to classify the water, oil and alcohol in honeycomb structure composites. Compared with LSTM and traditional 1D-CNN, the proposed model showed a better performance in automated liquid classification. The research topic is interesting. However, there are some points need to be addressed before consideration of publication.

1.      It is suggested to add some summary content of the sequential signal processing model in the current research.

2.      The LSTM network should be described in Section 3.3 rather than replaced by literature.

3.      Elaborate on the conclusion section on why the proposed algorithm is superior to other algorithms backed by some number from evaluation matrices.

Reviewer 2 Report

The study uses THz TDS in reflection to detect liquid ingress in honeycomb structures employed in aviation. An improved form of neural-network analysis is used to train the sensor system to identify ingress liquid.

Neural networks are not my area, so I cannot comment on that part of the paper. The parts describing THz measurements and data processing are technically correct and clearly explained.

If the neural-network parts are of a similar quality, the paper should be accepted.

Author Response

Thanks for your comments!

Reviewer 3 Report

1. Do you divide signals randomly into three sets? If no, then divide all signals into the training, validation, and testing sets randomly, and then run all experiments again.

2. The full meaning of loss, val_loss, acc, and val_acc need to be provided before using it in Figure 7.

3. The authors should use more evaluation metrics such as Accuracy, Sensitivity, Specificity, and Recall.

4. The authors should compare the proposed model with recent deep learning models and explain the total parameters and FLOPs.

5. Spectral classification is also an important classification method. The author should discuss it, such as the following paper.

Shen F, et al. Open-source mobile multispectral imaging system and its applications in biological sample sensing[J]. Spectrochimica Acta Part A: Molecular and Biomolecular Spectroscopy, 2022, 280: 121504.
